# Securing IoT Communications via Anomaly Traffic Detection: Synergy of Genetic Algorithm and Ensemble Method

**DOI:** 10.3390/s25134098

**Published:** 2025-06-30

**Authors:** Behnam Seyedi, Octavian Postolache

**Affiliations:** Department of Science, Instituto de Telecomunicacoes, ISCTE-University Institute of Lisbon, 1649-026 Lisbon, Portugal; seydi.behnam@gmail.com

**Keywords:** Internet of Things, genetic algorithm, XGBoost, anomaly detection, random forest

## Abstract

The rapid growth of the Internet of Things (IoT) has revolutionized various industries by enabling interconnected devices to exchange data seamlessly. However, IoT systems face significant security challenges due to decentralized architectures, resource-constrained devices, and dynamic network environments. These challenges include denial-of-service (DoS) attacks, anomalous network behaviors, and data manipulation, which threaten the security and reliability of IoT ecosystems. New methods based on machine learning have been reported in the literature, addressing topics such as intrusion detection and prevention. This paper proposes an advanced anomaly detection framework for IoT networks expressed in several phases. In the first phase, data preprocessing is conducted using techniques like the Median-KS Test to remove noise, handle missing values, and balance datasets, ensuring a clean and structured input for subsequent phases. The second phase focuses on optimal feature selection using a Genetic Algorithm enhanced with eagle-inspired search strategies. This approach identifies the most significant features, reduces dimensionality, and enhances computational efficiency without sacrificing accuracy. In the final phase, an ensemble classifier combines the strengths of the Decision Tree, Random Forest, and XGBoost algorithms to achieve the accurate and robust detection of anomalous behaviors. This multi-step methodology ensures adaptability and scalability in handling diverse IoT scenarios. The evaluation results demonstrate the superiority of the proposed framework over existing methods. It achieves a 12.5% improvement in accuracy (98%), a 14% increase in detection rate (95%), a 9.3% reduction in false positive rate (10%), and a 10.8% decrease in false negative rate (5%). These results underscore the framework’s effectiveness, reliability, and scalability for securing real-world IoT networks against evolving cyber threats.

## 1. Introduction

The rapid expansion of the Internet of Things (IoT) has revolutionized industries by enabling interconnected devices to enhance efficiency, automation, and innovation in sectors such as healthcare, industrial systems, and smart cities. This interconnectedness allows seamless communication and data exchange across heterogeneous devices, creating a dynamic ecosystem for IoT-driven advancements. However, IoT systems face significant security vulnerabilities due to their reliance on resource-constrained devices, decentralized architectures, and dynamic network environments. These vulnerabilities expose IoT ecosystems to cyber threats, such as anomalous network behavior, denial of service, and data manipulation attacks [1,2,3]. Machine learning (ML) has emerged as a promising approach to enhance IoT security by offering adaptive solutions for identifying anomalous behavior. Unlike traditional rule-based systems, ML techniques leverage patterns in IoT network behavior to detect deviations, enabling the identification of both normal and abnormal activities in real time [4,5]. Recent advancements in ML, including deep learning and feature extraction techniques, have significantly improved the precision and scalability of anomaly detection systems in IoT environments [6,7]. These systems identify known attack patterns and generalize them to detect previously unseen anomalies, making them suitable for dynamic IoT ecosystems [8,9]. Despite the promise of ML-based anomaly detection, several challenges remain. IoT networks generate large volumes of high-dimensional data, which can overwhelm traditional detection systems. Furthermore, the dynamic nature of IoT environments, characterized by frequent changes in device configurations and communication patterns, necessitates robust and adaptive models capable of maintaining detection accuracy [4,8]. Ensuring computational efficiency is another critical challenge, especially for IoT devices with limited processing power and memory [3,10]. Addressing these challenges requires the development of scalable, lightweight, and accurate anomaly detection frameworks that can operate effectively in resource-constrained and diverse IoT networks [2,5]. This study proposes a novel anomaly detection framework that distinguishes between normal and anomalous behavior in IoT networks. The proposed method balances scalability, accuracy, and resource efficiency by integrating advanced feature selection techniques, such as Genetic Algorithms, with ensemble machine learning classifiers like Random Forest, XGBoost, and Decision Tree models. Unlike traditional approaches, this framework generalizes well to diverse IoT environments and adapts dynamically to variations in network behavior.

In IoT environments, distinguishing between normal behavior and anomalous activities is essential for maintaining data integrity, confidentiality, and availability. While traditional intrusion detection systems focus on identifying specific types of attacks, they often fail to generalize and adapt to evolving network behavior. The primary challenge lies in designing a system that can accurately classify normal and anomalous behavior in various IoT scenarios without being limited to predefined attack patterns [4,6].

Although effective in detecting specific attack types, existing machine learning-based solutions struggle to handle the dynamic and heterogeneous nature of IoT networks [5]. Many approaches rely on exhaustive feature sets, leading to increased computational overhead and inefficiency in real-world deployments [7,10]. Moreover, a high false positive rate in anomaly detection can disrupt normal operations and reduce trust in the detection system. To address these limitations, it is critical to adopt feature selection techniques that optimize relevant features, reduce dimensionality, and enhance classification accuracy while maintaining computational efficiency [3,7]. The primary contributions of this study are outlined as follows:Novel Anomaly Detection Framework: This study proposes a robust three-phase anomaly detection framework for IoT environments. This framework integrates advanced data preprocessing, feature selection, and ensemble classification techniques to distinguish between normal and anomalous network behaviors effectively. By addressing the limitations of traditional intrusion detection systems, this method demonstrates adaptability and scalability across diverse IoT scenarios.Optimized Feature Selection: This study introduces a Genetic Algorithm-based feature selection technique, enhanced with optimization strategies inspired by nature (e.g., eagle-inspired search mechanisms). This approach identifies the most impactful features, reduces data dimensionality, and improves computational efficiency without compromising detection accuracy.Scalability in High-Risk Scenarios: This study examines the performance of the proposed method under varying attack percentages (η) and device risk levels (20% to 70%). This method maintains a high detection rate even in high-risk scenarios (η = 50%), showcasing its robustness and reliability for real-world IoT applications.

The remaining sections of this paper are the following: The Related Work section examines and analyzes prior studies and related works. The Materials and Methods section outlines the methodology’s key steps and components. The Results and Discussion section offers a comprehensive review of the simulation results. Finally, the Conclusion and Future Work section explores future research directions and concludes the paper.

## 2. Related Work

This study [11] uses deep learning techniques to propose an intrusion detection system for IoT networks. The model effectively detects diverse IoT attacks by leveraging the power of deep learning to identify anomalous behavior. The experimental results demonstrate high accuracy and scalability, making it a suitable solution for real-time IoT environments. The authors introduce an efficient feature extraction method to improve attack classification in IoT networks. This method ensures high accuracy in identifying various network intrusions by reducing computational complexity. Tests on IoT datasets confirm its ability to deliver precise and reliable classifications for enhanced security [12]. Tyagi et al. [13] explored supervised machine learning methods to detect attacks and anomalies in IoT networks. Through comprehensive experimentation, the models are shown to achieve high accuracy and sensitivity. These methods are especially effective in addressing the growing diversity of IoT network threats. In [14], a robust attack detection framework using ensemble classifiers is presented, explicitly targeting industrial IoT (IIoT) environments. This study demonstrates the approach’s ability to handle imbalanced datasets and maintain high detection accuracy. It proves to be a reliable solution for real-world IIoT security challenges.

The authors of [15] designed a machine learning-based intrusion detection framework tailored to IoT networks. The proposed methodology adapted well to complex scenarios, delivering high accuracy and resilience. Extensive experiments validated the framework, proving its effectiveness for dynamic IoT environments. Zhao et al. [16] presented a novel intrusion detection system using a lightweight neural network designed for the IoT. It achieved high detection accuracy while minimizing resource consumption, making it ideal for resource-constrained IoT devices. The results validated its potential for efficient and scalable IoT security. The authors of [17] proposed an intrusion detection mechanism using autoencoders and data partitioning to enhance efficiency in IoT networks. This approach significantly reduced computational overhead while achieving high detection precision. Their methodology suits dynamic IoT environments with large-scale data. An AI-based anomaly detection scheme was introduced for identifying cyber threats in IoT-based transport networks. The method excelled in proactive threat prevention and offers robust performance in securing IoT-enabled transport systems. This highlights the role of AI in advancing IoT cybersecurity [18].

The study in [19] presents a two-level hybrid model for detecting anomalous activities in IoT networks. The model integrates statistical and machine learning methods to improve accuracy and reliability. Its experimental results indicate enhanced performance in identifying diverse IoT anomalies while maintaining low false positive rates, making it practical for real-world applications. Huang et al. [20] proposed an Energy-efficient and Trustworthy Unsupervised (EATU) anomaly detection framework for the industrial IoT (IIoT). Their framework ensures reliable anomaly detection while minimizing resource usage by focusing on unsupervised learning and energy efficiency. Their study’s empirical evaluation demonstrated that EATU proves effective in dynamic and resource-constrained IIoT environments. Altulaihan et al. [21] introduced an anomaly detection intrusion detection system (IDS) to identify DoS attacks in IoT networks. The system employed machine learning algorithms to ensure high detection accuracy and robustness. The results confirmed its capability to address critical IoT security threats, focusing on detecting DoS attacks. Another paper proposed a two-tier hybrid ensemble learning pipeline for intrusion detection in IoT networks. By combining multiple machine learning techniques, the pipeline significantly improved detection accuracy and efficiency. The approach was validated on IoT datasets, showcasing its potential for enhancing network security [22].

The study in [23] presents an intrusion detection system (IDS) for IoT networks that utilizes an ensemble of unsupervised techniques. The framework effectively addresses the challenges of diverse intrusion types and imbalanced datasets. Experimental evaluations highlight its high performance in identifying intrusions while reducing computational complexity. The authors propose EIDM, a deep learning-based model for intrusion detection in IoT networks. The model is designed to detect various intrusions effectively using advanced deep learning techniques. Experimental evaluations demonstrate EIDM’s high accuracy and robustness in handling complex IoT datasets. Additionally, the model showcases its scalability and suitability for real-world IoT environments, making it a promising solution for enhancing IoT network security [24]. Mohapatra et al. [25] addressed handling MITM attacks in wireless sensor networks (WSNs) through IDS. Their approach provided a framework to identify and mitigate MITM attacks, improving the security of sensor-based applications. Recent research has increasingly applied graph neural networks (GNNs) to model complex transportation and infrastructure systems. Liu and Meidani [26] introduced an end-to-end heterogeneous GNN for user-equilibrium traffic assignment, which leverages virtual origin destination links and graph-based flow conservation to improve prediction accuracy and adaptability across varying network topologies. In another work [27], a GNN-based surrogate model was proposed for the rapid seismic reliability analysis of highway bridge systems, effectively capturing node-level connectivity patterns under uncertainty. These GNN-based approaches demonstrate the potential of learning from graph-structured data in dynamic and complex networks. Table 1 summarizes the advantages and disadvantages of the related work.

Despite significant advancements in machine learning-based intrusion detection systems (IDSs) for IoT networks, several critical gaps remain in the existing research. Deep learning-based approaches [1,6,14] demonstrate high detection accuracy but suffer from substantial computational demands, making them unsuitable for resource-constrained IoT devices. Feature extraction and selection methods [2,7,17] aim to reduce complexity but often compromise detection accuracy or fail to generalize across diverse environments. Supervised machine learning methods [3,5,16] and ensemble classifiers [4,12,13] enhance performance yet frequently overlook dynamic adaptation to heterogeneous and evolving IoT network conditions. The proposed three-phase framework addresses these limitations by introducing the following: (1) an advanced data preprocessing phase using the Median-KS Test and robust techniques for noise elimination, missing value imputation, and dataset balancing, ensuring cleaner and more structured input data; (2) a feature selection and hyperparameter optimization phase leveraging a Genetic Algorithm enhanced with eagle-inspired search behavior and simulated annealing improved via dragonfly swarm dynamics, achieving an optimal trade-off between dimensionality reduction and classification accuracy; and (3) a classification phase employing a powerful ensemble of Decision Tree, Random Forest, and XGBoost, ensuring scalability, robustness, and adaptability across dynamic IoT scenarios. This holistic approach enhances detection performance while minimizing computational costs, making it highly effective for real-world, resource-constrained, and heterogeneous IoT environments.

## 3. Materials and Methods

The proposed approach consists of three main phases, each playing a key role in enhancing the system’s performance using advanced algorithms. In the first phase, data preprocessing techniques such as the Median-KS Test are employed to eliminate noise and identify valid data. Additionally, advanced methods are applied to handle missing values, balance datasets, and prepare categorical features, transforming raw data into an analyzable format. In the second phase, feature selection and optimization, a Genetic Algorithm (GA) enhanced with eagle-inspired search strategies is employed to identify and select the most influential features, effectively reducing the dimensionality of the data while maintaining high classification performance. In parallel, hyperparameter tuning is performed using a simulated annealing (SA) algorithm, which is further improved by integrating dragonfly-inspired swarm behaviors (alignment, cohesion, separation) to enhance exploration and exploitation capabilities during the search for optimal model parameters. Finally, a combination of three machine learning algorithms, Decision Tree, Random Forest, and XGBoost, is used to classify the data. This three-phase structure offers a comprehensive and practical approach to detecting and countering security threats in the IoT environment. A flowchart of the proposed method is shown in Figure 1.

### 3.1. Phase 1—Preprocessing

The preprocessing phase prepares the data for further analysis by addressing common issues such as missing values (NaN), categorical features, imbalanced datasets, and missing entries. This step is critical for ensuring the quality and consistency of the data, enabling better downstream analysis and minimizing obstacles that could disrupt the data processing flow. Handling NaN values is one of the first steps in preprocessing. NaN, short for “Not a Number”, is commonly used to represent missing data in datasets. If left unaddressed, these values can negatively impact the accuracy and reliability of the attack detection system. To ensure high precision in the results, all NaN values must be appropriately handled and replaced with suitable substitutes. Managing NaN values in the input data involves systematic cleaning to ensure the dataset is ready for effective processing. This stage sets the foundation for subsequent phases by creating a cleaner, more consistent dataset that enhances the reliability of the analytical models. This process can be described in (1) as follows:(1)$$d_{NaN\text{-}handled}=M_{NaN}(d_{input})$$
where $d_{input}$ represents the raw dataset, $M_{NaN}$ is the method used to handle NaN values, and $d_{Nan\text{-}handled}$ is the dataset after addressing NaN values.

After addressing NaN values, the dataset often contains categorical features and non-numerical attributes like labels or text descriptions. Since machine learning models are inherently mathematical, categorical data must be transformed into numerical formats to ensure compatibility. Techniques such as one-hot encoding, label encoding, and ordinal transformation are commonly used. This process can be described as (2):(2)$$d_{cat\text{-}handled}=M_{cat}(d_{input})$$
where $d_{input}$ represents the raw dataset, $M_{cat}$ is the transformation method, and $d_{cat\text{-}handled}$ is the dataset after processing categorical features.

Missing data present another challenge during preprocessing. These missing values may occur randomly across data samples or follow a structured pattern. Random missing values, known as “missing at random”, are distributed without any clear correlation, while structured missing values, or “missing not at random”, follow specific patterns that might correlate with other features. After analyzing the dataset, we observed that missing values were mainly present in certain features such as packet-related statistics. To address this, we applied mean imputation for continuous features and mode imputation for categorical features. Additionally, samples with excessive missing data (>30% missing attributes) were removed to preserve the dataset’s quality. Managing these scenarios involves tailored strategies, represented as (3):(3)$$d_{mis\text{-}handled}=M_{mis}(d_{input})$$
where $d_{input}$ represents the raw dataset, $M_{mis}$ denotes the method to address missing values, and $d_{mis\text{-}handled}$ is the dataset after handling missing values. Once issues like NaN values, categorical features, and missing data are addressed, the preprocessed dataset is assembled. The final dataset is expressed as (4):(4)$$d_{preprocessed}=\{d_{1},d_{2},\ldots ,d_{N}\}$$
where $d_{preprocessed}$ is the preprocessed dataset and N represents the total number of data samples.

### 3.2. Phase 2—Feature Extraction

The feature extraction phase builds on the balanced data to identify patterns and insights critical for detecting abnormalities. A novel method, the Median Absolute Deviation around the Median-based Kolmogorov–Smirnov Test [28] (MAD Median-KS Test), was developed to achieve this. This method incorporates statistical analysis techniques and leverages data from the network to detect anomalies caused by compromised IoT devices. The MAD Median-KS Test enhances traditional approaches by focusing on distribution similarity and removing redundant data. This is unlike the conventional Kolmogorov–Smirnov (KS) test, which emphasizes the largest difference in distribution, making it susceptible to outliers and reducing its reliability in attack detection. This improved method accounts for median absolute deviations around the median to mitigate the influence of outliers. The test computes the features of the input data using Equation (5):(5)$$Features=\lambda \cdot \Gamma_{n}(\partial (D)-\Gamma_{i}(\tau (D)))$$
where λ is a constant used to filter abnormalities caused by outliers. $\Gamma_{n}$ represents the median of the distribution functions. ∂ and τ are two distribution functions derived from the balanced dataset D. $\Gamma_{i}$ denotes the median of τ, capturing key characteristics of the data.

#### Feature Selection

The Genetic Algorithm (GA) is a widely recognized evolutionary optimization technique that simulates natural selection, where the fittest individuals are chosen to pass their traits to the next generation. In feature selection, GA is utilized to identify the most relevant subset of features that significantly improve model performance while discarding irrelevant or redundant ones. This approach minimizes the dimensionality of the dataset, enhancing computational efficiency without sacrificing accuracy. A novel component of this process is the incorporation of optimization techniques inspired by the hunting behavior of North American bald eagles. These eagles select areas rich in food and execute efficient spiral movements to capture prey. This behavior is mathematically represented to guide the optimization process. The initial position of a search is calculated using (6):(6)$$X_{n,i}=X_{b}+\eta \cdot \delta \cdot (X_{avg}-X_{i})$$
where $X_{b}$ is the best location identified during the search, $X_{avg}$ is the average position, $X_{i}$ is the current position, η controls positional variations, and δ is a binary variable r ∈ {0, 1}. After identifying a promising location, the eagle refines its search through a spiral movement defined by (7):(7)$$X_{i,n}=X_{i}+y_{i}\times (X_{i}-X_{i+1})+z_{i}\times (X_{i}-X_{m})$$
where(8)$$z_{i}=\frac{z\delta (i)}{\max|z\delta |}$$(9)$$y_{i}=\frac{y\delta (i)}{\max|y\delta |}$$(10)zδ(i)=δ(i)×sin(θ(i))(11)yδ(i)=δ(i)×cos(θ(i))(12)θ(i)=ϕ×π×rand(13)δ(i)=θ(i)+ω×rand

In this model, θ(i) represents the angle of search and ω denotes the number of search cycles (0.5 ≤ ω ≤ 2). During the swooping phase, the eagle targets its prey by diving into an optimal location. This behavior is modeled as follows:(14)$$X_{i,n}=rand\cdot X_{b}+z_{1}(i)\times (X_{i}-C_{1}\times X_{m})+y_{1}(i)\times (X_{i}-C_{2}\times X_{b})$$
where $C_{1}$ and $C_{2}$ are control coefficients, and(15)$$z_{1}(i)=\frac{z\delta (i)}{\max|z\delta |}$$(16)$$y_{1}(i)=\frac{y\delta (i)}{\max|y\delta |}$$(17)zδ(i)=δ(i)×sinh(θ(i))(18)yδ(i)=δ(i)×cosh(θ(i))(19)θ(i)=ϕ×π×rand

The GA-based feature selection algorithm incorporates a fitness function that evaluates the quality of feature subsets. Fitness is calculated as follows:(20)$$Fitness=w_{1}\cdot L(f)+w_{2}*(1-\frac{\left|S\right|}{\left|T\right|})$$
where L(f) is the classification error for the feature subset f, ∣S∣ is the count of selected features, ∣T∣ is the total number of features in the dataset, and $w_{1}$ and $w_{2}$ are weighting parameters with $w_{1}+w_{2}=1$. The fitness function defined in Equation (20) is designed to optimize two main objectives: minimizing classification errors and reducing model complexity by selecting the most relevant features. The first term, L(f), measures the classification error for a given feature subset. This ensures that the selected features contribute significantly to improving the model’s performance. The second term, |S|/|T|, represents the ratio of selected features to the total available features. This term is introduced to prevent the selection of excessive and redundant features, which can increase computational complexity and reduce efficiency, especially in resource-constrained environments such as IoT networks. The coefficients $w_{1}$ and $w_{2}$ act as weighting parameters that control the importance of each term. Their values are carefully tuned based on the problem’s requirements to maintain a balance between classification accuracy and computational efficiency.

### 3.3. Phase 3—Classification

The proposed approach integrates an ensemble voting classifier [29] to enhance the effectiveness of a network intrusion detection system (NIDS). This ensemble method combines the strengths of multiple ML algorithms: XGBoost [30], Random Forest (RF) [31], and Decision Tree (DT) [32]. By leveraging their unique capabilities, the ensemble classifier achieves superior accuracy and robustness in detecting malicious traffic.

#### 3.3.1. Decision Tree

A Decision Tree is a machine learning model structured like a tree for classification or regression tasks. It recursively splits the dataset based on specific features to create smaller data groups that are more homogeneous. Each internal node in the tree represents a decision based on a feature, and each leaf node corresponds to a final prediction or outcome. The splitting criterion typically uses a cost function such as GINI Impurity or Entropy in classification trees. The formula for Entropy is given by the following:(21)$$Entropy=-\sum_{i=1}^{k}p_{i}\cdot \log p_{i}$$
where $p_{i}$ is the probability of class i.

#### 3.3.2. Random Forest

Random Forest is an ensemble learning algorithm that combines multiple decision trees to improve accuracy and reduce overfitting. It generates a set of decision trees, each trained on a randomly selected subset of data and features. For classification tasks, it uses majority voting across the trees to determine the final prediction, and for regression tasks, it averages the predictions of the trees. A key feature of Random Forest is Bootstrap Aggregation (Bagging), which involves sampling the data with a replacement to create diverse training subsets. This technique helps reduce overall error and increase model robustness.

#### 3.3.3. XGBoost

XGBoost, short for “Extreme Gradient Boosting”, is an advanced ensemble algorithm that builds upon gradient boosting frameworks to efficiently handle regression and classification problems. It trains decision trees sequentially, where each tree attempts to correct the errors of the previous ones. XGBoost incorporates regularization to prevent overfitting and parallel processing for faster training and handling missing values. The loss function in XGBoost is optimized using gradient descent, and a second-order Taylor expansion of the loss function is often used to enhance convergence and stability:(22)$$Loss\approx \sum_{i=1}^{n}\left[g_{i}\cdot h(x_{i})+\frac{1}{2}h_{i}\cdot h{\left(x_{i}\right)}^{2}\right]$$
where $g_{i}$ is the gradient and $h_{i}$ is the Hessian for the loss function. XGBoost is highly efficient and widely used in machine learning competitions for its performance.

#### 3.3.4. Ensemble Voting Classifier

The ensemble voting classifier combines predictions from multiple base models to improve the accuracy of the final classification. Two types of voting mechanisms are employed:

Voting 1: The final class label is determined by the majority vote of the base models.(23)$$C=majority\;voting(C_{1},C_{2},\ldots ,C_{n})$$

Voting 2: Probability estimates from each base model are averaged, and the class label with the highest probability is selected.(24)$$C=\underset{c}{\arg\max}\sum_{i=1}^{n}P_{i,c}$$
where C is the final class label, and $P_{i,c}$ is the probability of class c predicted by the i-th base model.

### 3.4. Hyperparameter Tuning

Simulated annealing (SA) was utilized in this study as a parameter-tuning method. SA is inspired by the natural process of annealing in metallurgy, where materials are heated and cooled gradually to remove defects and optimize their structure. In the context of this study, SA was enhanced by incorporating inspiration from swarm intelligence behaviors observed in dragonflies, particularly to improve its exploration and exploitation abilities. It is important to note that the complete Dragonfly Algorithm (DA) was not used. The dragonfly-inspired approach incorporates five key behavioral factors: alignment, which adjusts an individual’s position based on the average location of its neighbors; attraction, which guides movement toward desired targets; cohesion, which minimizes positional variance within the group; separation, which prevents overcrowding; and distraction, which helps avoid undesirable or “enemy” locations. These factors collectively influence the movement and positioning of individuals within the swarm to identify optimal solutions. The SA algorithm updates the position of each iteratively using a mathematical framework. This includes calculating a step vector that combines contributions from alignment, attraction, cohesion, separation, and distraction. When no neighbors are present within a defined radius, a random walk based on the Levy flight process is employed to maintain exploration of the search space. The algorithm dynamically adjusts its parameters such as alignment, cohesion, and neighborhood radius to balance global exploration in early iterations with local refinement in later stages. The fitness function used in SA evaluates the classification error rate, defined as the percentage of misclassified instances relative to the total number of instances. The goal is to minimize this error rate, ensuring the model achieves high accuracy and generalization while avoiding overfitting. By iteratively fine-tuning hyperparameters in this manner, our SA-based approach enables a more efficient and effective exploration of the parameter space compared to traditional tuning methods such as grid search or random search.(25)$$Fitness=\frac{Number\;of\;misclassified}{Total\;number\;of\;instances}\times 100$$

## 4. Results and Discussion

In this section, the proposed method is compared with four other methods, namely Support Vector Machine (SVM), Convolutional Neural Network (CNN), Ensemble Neural Networks (ENNs), and Deep Belief Network (DBN). The architecture and hyperparameter settings of the baseline models are summarized in Table 2.

The simulation results of all methods are obtained on the CTU-13 dataset [33]. The dataset is inherently imbalanced, consisting of 4775 botnet instances and 20,902 normal instances. For model development, 80% of the data was allocated for training, while the remaining 20% was used for testing. The CTU-13 dataset, compiled by the Czech Technical University, is a widely used benchmark for evaluating intrusion detection systems, especially in IoT network security research. It contains thirteen different capture scenarios of real botnet traffic mixed with normal and background traffic. Each scenario includes various types of botnet activities such as spam, click fraud, and DDoS attacks, providing a realistic environment for anomaly detection. We selected the CTU-13 dataset because of its diversity in attack behaviors, the complexity of real network traffic, and its relevance to modern IoT environments where devices communicate dynamically across heterogeneous networks. Furthermore, its widespread use in recent research [33] allows for fair comparisons with existing methods while validating the robustness and generalization capability of our proposed anomaly detection framework.

The performance evaluation of the proposed method is carried out using the following key metrics: accuracy, specificity, sensitivity, precision, F-measure, false positive rate (FPR), false negative rate (FNR), Matthews Correlation Coefficient (MCC), and detection rate. Below is a detailed explanation of each metric:

Accuracy: This metric indicates the proportion of correctly identified instances (positive and negative) out of the total instances. A high accuracy demonstrates the overall reliability of the detection system (26).(26)$$Accuracy=\frac{TP+TN}{TP+TN+FP+FN}$$

Specificity: Specificity measures the system’s ability to correctly identify normal (negative) cases. A higher specificity reflects fewer false alarms and the more precise detection of attack-free scenarios (27).(27)$$Specificity=\frac{TN}{TN+FP}$$

Sensitivity (Recall): Sensitivity quantifies the system’s ability to detect attack (positive) cases accurately. A high sensitivity value effectively identifies malicious activities (28).(28)$$Sensitivity=\frac{TP}{TP+FN}$$

Precision is the ratio of true positives to the sum of true and false positives. It reflects the system’s accuracy in identifying attack cases without misclassifying normal instances (29).(29)$$Precision=\frac{TP}{TP+FP}$$

F-Measure: This metric provides a harmonic mean of precision and sensitivity, offering a balanced evaluation of the system’s performance, especially when class distribution is uneven (30).(30)$$F\;measure=2\times \frac{Precision\times Sensitivity}{Precision+Sensitivity}$$

False Positive Rate (FPR): FPR measures the proportion of normal instances incorrectly classified as attacks. A lower FPR highlights the system’s ability to avoid unnecessary false alarms (31).(31)$$FPR=\frac{FP}{FP+TN}$$

False Negative Rate (FNR): FNR quantifies the proportion of attack instances incorrectly classified as normal. A lower FNR ensures effective attack detection without overlooking threats (32).(32)$$FNR=\frac{FN}{FN+TP}$$

Matthews Correlation Coefficient (MCC): MCC provides a balanced evaluation of the system’s predictions, accounting for true and false positives and negatives. It is particularly useful for imbalanced datasets (33).(33)$$MCC=\frac{\left(TP\times TN)-(FP\times FN\right)}{\sqrt{\left(TP+FP)(TP+FN)(TN+FP)(TN+FN\right)}}$$

Detection Rate: This metric represents the ratio of successfully detected attack cases to the total number of attack instances. A higher detection rate indicates the system’s robustness in identifying attacks across varying scenarios (34).(34)$$DetectionRate=\frac{TP}{TP+FN}$$

### 4.1. Simulation Results

To evaluate the optimization performance of the proposed and baseline feature selection methods, we analyze the fitness values obtained at different iteration steps. Table 3 presents the convergence behavior of several metaheuristic algorithms, including the Chimp Optimization Algorithm (ChOA) [34], Particle Swarm Optimization (PSO) [35], Social Spider Optimization (SSO) [36], Sparrow Search Algorithm (SPA) [37], and proposed feature selection method. Fitness values are reported at iterations 10 through 50, showing that the proposed method consistently achieves faster convergence and higher final fitness values compared to the other approaches.

In Table 3 are presented the values associated with the performance improvement (e.g., objective function values) at each iteration. The results indicate that the proposed feature selection method achieves faster and more significant convergence than the baseline techniques. The proposed method consistently outperforms the others at each iteration, reaching higher values at earlier iterations. For instance, after 10 iterations, the proposed method performs 129, significantly higher than ChOA (110) and PSO (86). Similarly, by the 50th iteration, the proposed method achieves a value of 210, surpassing ChOA (188) and the other techniques. This superior convergence behavior can be attributed to the advanced optimization strategies integrated into the proposed method, such as adaptive parameter tuning and enhanced exploration and exploitation capabilities. These features enable the method to navigate the solution space more effectively, avoiding local optima and accelerating convergence. The ability to converge faster and achieve higher performance demonstrates the proposed method’s robustness and efficiency, making it a more reliable choice for feature selection in high-dimensional datasets. Figure 2 compares five classification models: ENN, CNN, SVM, DBN, and the proposed method. The metrics used for evaluation include accuracy, specificity, sensitivity, and precision, displayed in subplots (a), (b), (c), and (d), respectively. The superior performance of the proposed method can be attributed to its sophisticated and well-optimized structure. Unlike traditional models, the proposed method incorporates advanced preprocessing, feature selection, and classification strategies. During preprocessing, noise and outliers are effectively managed using techniques such as the Median-KS Test. The feature selection phase employs Genetic Algorithms, which reduce dimensionality and optimize the feature set, ensuring the inclusion of highly relevant attributes. Finally, in the classification phase, an ensemble voting classifier combines the strengths of the Decision Tree, Random Forest, and XGBoost algorithms, significantly improving prediction accuracy. These enhancements enable the proposed method to handle complex patterns more efficiently and reduce false positives and negatives, achieving a balanced performance across all evaluation metrics.

Figure 3 compares five classification models: ENN, CNN, SVM, DBN, and the proposed method. The metrics used for evaluation include the F-measure, FPR, FNR, and MCC, displayed in subplots (a), (b), (c), and (d), respectively. The proposed method minimizes noise and irrelevant attributes by effectively addressing data inconsistencies during preprocessing and utilizing Genetic Algorithms to select the most impactful features. Additionally, the ensemble voting classifier, which combines the strengths of the Decision Tree, Random Forest, and XGBoost models, ensures high accuracy, robust classification, and scalability. The proposed method’s ability to reduce FPR and FNR is critical for real-world applications, where minimizing false alarms and undetected threats is essential. Moreover, its superior MCC scores reflect the proposed method’s capability to maintain consistent performance across balanced and imbalanced datasets, further emphasizing its reliability and adaptability in various scenarios.

To evaluate the robustness of the proposed method under various threat levels, training datasets with different attack percentages (20%, 30%, 40%, and 50%) were generated. This was achieved by adjusting the ratio of botnet to normal traffic using a combination of random under-sampling (for majority class) and over-sampling (for minority class) strategies on the training set without altering the test set. Due to the inherent imbalance in the CTU-13 dataset, the Synthetic Minority Over-sampling Technique (SMOTE) was applied on the training data to generate synthetic samples for the botnet class. This ensured a more balanced dataset, improved the model’s ability to learn minority class patterns, and enhanced overall classification performance. Figure 4 illustrates the detection rate performance under varying attack percentages (η) and different proportions of at-risk devices (from 20% to 70%). Subplots (a), (b), (c), and (d) correspond to attack percentages of η = 20%, η = 30%, η = 40%, and η = 50%, respectively.

The proposed method consistently achieves the highest detection rates across all scenarios and percentages of devices at risk, showcasing its robustness and superior anomaly detection capability. Even as the percentage of devices at risk increases (from 20% to 70%), the detection rate of the proposed method declines more gracefully compared to the other methods, demonstrating its ability to handle high-risk conditions effectively. ENN and CNN show moderate detection rates but fail to match the stability and effectiveness of the proposed method as the percentage of devices at risk increases. SVM and DBN exhibit more significant declines in their detection rate with increasing devices at risk, particularly in high-risk scenarios (η = 50). The gap between the proposed method and the other models widens as η and the percentage of devices at risk increases, further emphasizing the robustness of the proposed approach. The superior performance of the proposed method can be attributed to its advanced feature selection and classification strategies. The proposed method ensures better generalization and adaptability by integrating techniques like the Genetic Algorithm for optimal feature selection and ensemble classifiers for improved prediction accuracy. Its capability to maintain a high detection rate under challenging conditions stems from the ensemble voting mechanism, which combines the strengths of multiple models (Decision Tree, Random Forest, and XGBoost). This multi-layered approach enables the proposed method to detect complex patterns and minimize false positives and negatives, ensuring consistent performance even in high-risk scenarios.

Figure 5 illustrates the confusion matrices for training and testing phases at varying percentages of data utilization. In the 80% training phase, the method achieves remarkable results with only 27 false positives and 60 false negatives, indicating high precision and sensitivity in detecting botnet activity. Similarly, the model maintains consistent performance in the 70% training phase, with a minimal increase in false positives (28) and a slight reduction in false negatives (33). This demonstrates the robustness of the proposed approach in accurately identifying botnet and normal traffic, even with reduced training data. During the 20% testing phase, the confusion matrix shows only 7 false positives and 20 false negatives, reflecting the model’s high generalization capability and ability to accurately predict botnet instances under unseen data conditions. The model performs exceptionally well in the 30% testing phase, with just 12 false positives and 50 false negatives. This indicates the scalability and reliability of the proposed method in handling larger test datasets.

In Table 4 is presented an evaluation of the proposed method performance in comparison with several existing botnet detection approaches across different datasets. The comparison focuses on key performance metrics, namely accuracy, detection rate, and false positive rate (FPR). The proposed method achieves an accuracy of 98.0%, detection rate of 95.0%, and low FPR of 7.8% on the CTU-13 dataset. Compared to Sharma and Babbar (2024) [33], who also utilized the CTU-13 dataset but achieved lower accuracy (95.2%) and a significantly higher FPR (18.0%), the proposed framework demonstrates enhanced effectiveness. Furthermore, although Elnakib et al. (2023) [24] and Ullah and Mahmoud (2019) [19] reported competitive performances on Bot-IoT and N-BaIoT datasets, respectively, their results indicated lower detection rates and higher false positive rates compared to the proposed method. These findings confirm the robustness, scalability, and generalization capability of our proposed three-phase anomaly detection framework in securing IoT networks against diverse and sophisticated botnet attacks.

To ensure a robust evaluation, we adopted a five-fold cross-validation scheme. The dataset was randomly partitioned into five equally sized folds, and each fold was used once as a test set while the remaining four were used for training. We reported the average and standard deviation of three standard classification metrics: accuracy, detection rate, and FPR. These metrics were chosen to reflect both overall performance and anomaly detection capability, especially in the presence of class imbalance. The results in Table 5 demonstrate the stability and reliability of the proposed framework across multiple runs.

To determine the best ensemble strategy, both Hard Voting and Soft Voting mechanisms were evaluated. As presented in Table 6, Soft Voting achieved a higher accuracy (98.0%) and detection rate (95.0%) while maintaining a lower false positive rate (7.8%) compared to Hard Voting. Therefore, Soft Voting was selected as the final voting strategy in the proposed anomaly detection framework.

### 4.2. Discussion

The superior performance of the proposed GA-ensemble framework can be attributed to its synergistic design. The Genetic Algorithm, enhanced with eagle-inspired search behavior, ensures optimal feature selection by exploring the search space more effectively and avoiding local minima. This reduces dimensionality without compromising accuracy. Furthermore, the dragonfly-inspired simulated annealing algorithm refines hyperparameters in a biologically inspired manner, enhancing model generalization. The ensemble classifier integrates diverse models, namely Decision Tree, Random Forest, and XGBoost, each contributing complementary strengths. The combination of Decision Tree, Random Forest, and XGBoost provides both a bias–variance trade-off and resilience to imbalanced data.

The proposed framework offers several practical advantages for securing real-world IoT environments. Its modular design, combining lightweight preprocessing, nature-inspired feature selection, and an ensemble of efficient classifiers, makes it suitable for deployment in edge-based applications such as smart homes, healthcare monitoring devices, the industrial IoT (IIoT), and autonomous sensor networks. The low false positive and false negative rates ensure minimal disruption to normal system operations, which is critical in mission-critical IoT deployments. The strengths of the model include its high detection accuracy (98%), generalization to diverse attack types, and robustness under varying attack ratios and device risk levels. The ensemble learning strategy combined with adaptive optimization techniques ensures a balance between computational efficiency and detection performance. However, the model has two notable limitations: its performance may degrade when dealing with encrypted or heavily obfuscated traffic, as the current design relies primarily on flow-level statistical features; and despite using automated tuning via swarm-enhanced simulated annealing, the training phase still incurs a degree of computational overhead that must be considered in resource-limited deployment scenarios.

## 5. Conclusions

The proposed anomaly detection framework for IoT networks effectively addresses critical security challenges by detecting anomalous behaviors and preventing cyber threats in dynamic and resource-constrained environments. By integrating advanced data preprocessing, feature selection through a Genetic Algorithm enhanced with eagle-inspired search mechanisms, and an ensemble classification approach, the framework is characterized by its high performance expressed through its accuracy, scalability, and computational efficiency. The obtained results demonstrate significant improvements compared to existing methods. The framework achieves a 12.5% increase in accuracy (98%), a 14% improvement in detection rate (95%), a 9.3% reduction in false positive rate (10%), and a 10.8% decrease in false negative rate (5%). These improvements clearly highlight the framework’s effectiveness and superiority in ensuring IoT security. Its notable enhancement in accuracy is a direct result of the optimized feature selection and ensemble learning approach, which collectively contribute to more precise and reliable anomaly detection. Future research can focus on improving the scalability and real-time deployment of the proposed framework to handle large-scale IoT networks with thousands of interconnected devices while ensuring minimal latency and high adaptability. Additionally, enhancing the framework’s capability to detect complex, multi-stage attacks such as Advanced Persistent Threats (APTs) can significantly improve its effectiveness in real-world cybersecurity scenarios. Another promising direction is the integration of federated learning techniques, which would enable decentralized model training while preserving user privacy, making the system more secure and practical for IoT applications that prioritize data confidentiality.

## Figures and Tables

**Figure 1 sensors-25-04098-f001:**
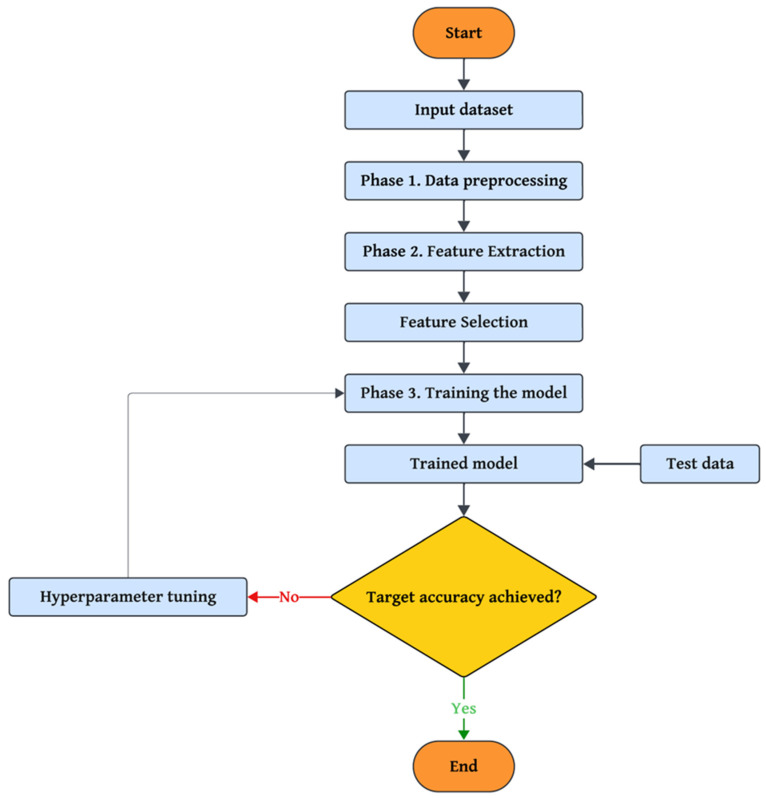
Flowchart of the proposed method.

**Figure 2 sensors-25-04098-f002:**
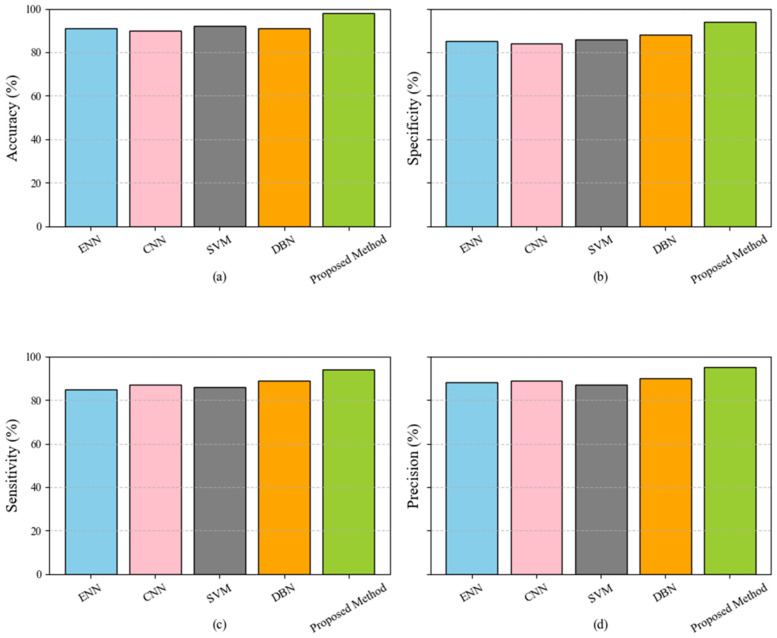
Performance comparison of classification models (ENN, CNN, SVM, DBN, and proposed method) across evaluation metrics: accuracy (**a**), specificity (**b**), sensitivity (**c**), and precision (**d**).

**Figure 3 sensors-25-04098-f003:**
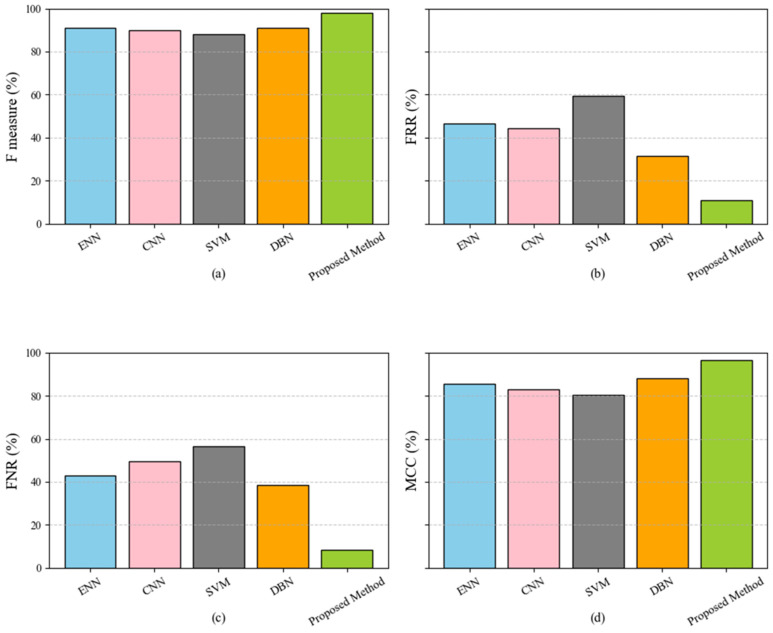
Performance comparison of classification models (ENN, CNN, SVM, DBN, and proposed method) across evaluation metrics: F-measure (**a**), FPR (**b**), FNR (**c**), and MCC (**d**).

**Figure 4 sensors-25-04098-f004:**
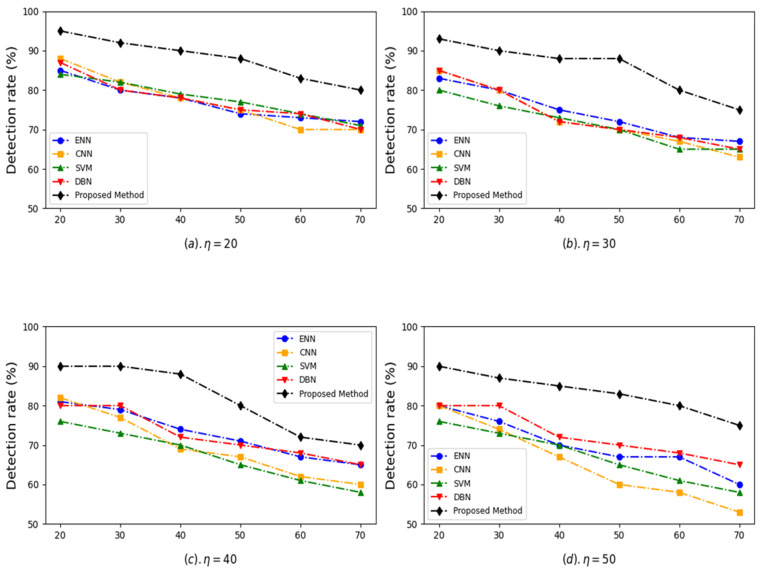
Detection rate comparison of ENN, CNN, SVM, DBN, and proposed method across different attack percentages.

**Figure 5 sensors-25-04098-f005:**
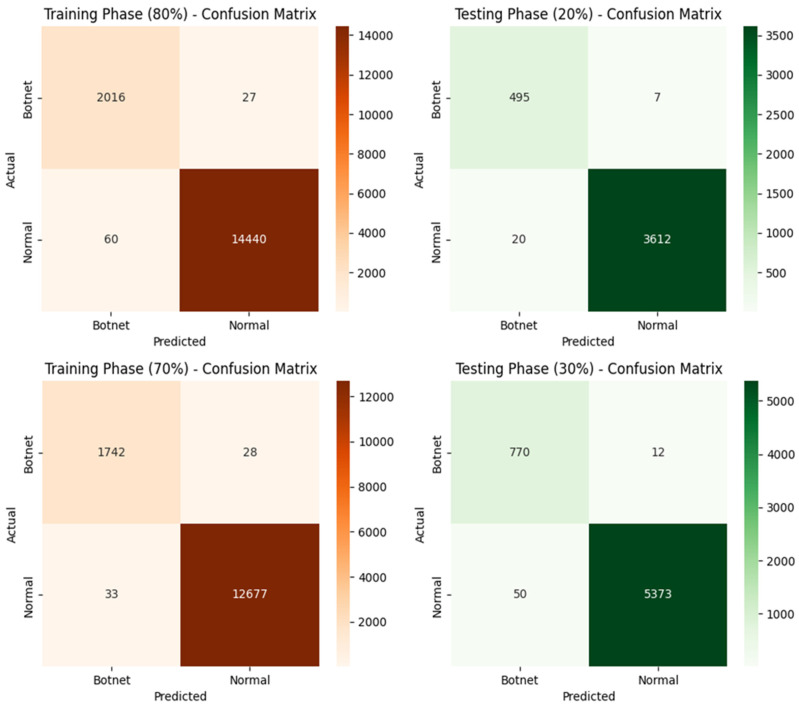
Confusion matrices for both training and testing phases.

**Table 1 sensors-25-04098-t001:** Advantages and disadvantages of related work.

Ref.	Year	Advantages	Disadvantages
[11]	2023	High accuracy and scalability for diverse IoT attack detection.	Potential high computational demands for deep learning.
[12]	2021	Efficient feature extraction with reduced computational complexity.	May require further tuning for diverse IoT scenarios.
[13]	2021	Effective in addressing diverse IoT threats with high accuracy and sensitivity.	Limited evaluation of highly dynamic IoT environments.
[14]	2021	Handles imbalanced datasets and maintains high detection accuracy.	Performance may depend on the quality of the ensemble models.
[15]	2022	Adapts well to complex scenarios with high performance and reliability.	Framework may require additional resources for real-time scenarios.
[16]	2021	High detection accuracy and low resource consumption for constrained devices.	Lightweight models may miss subtle intrusion patterns.
[17]	2024	Reduces computational overhead and improves detection precision in large-scale environments.	Effectiveness may vary in non-IoT environments.
[18]	2023	Proactive threat prevention with robust performance in IoT-based transport networks.	Limited focus on broader IoT environments beyond transport systems.
[19]	2019	Improved anomaly detection accuracy with low false positives.	A hybrid approach could increase system complexity.
[20]	2022	Energy-efficient and trustworthy detection suitable for IIoT environments.	Reliance on unsupervised learning may limit detection specificity.
[21]	2024	High detection accuracy and robustness for detecting DoS attacks.	Focuses primarily on DoS attacks; may need an extension for other threats.
[22]	2023	Enhanced detection accuracy and efficiency with hybrid ensemble learning.	A complex pipeline may increase implementation overhead.
[23]	2022	Effectively identifies diverse intrusion types with reduced computational complexity.	May require fine-tuning for specific IoT environments.
[24]	2023	High accuracy and robustness in handling complex IoT datasets.	A deep learning model may demand significant computational resources.
[25]	2020	IDS approach for MITM attack handling in wireless sensor networks.	May struggle with scalability in large-scale WSNs or with high data traffic.

**Table 2 sensors-25-04098-t002:** Architecture and hyperparameter settings of baseline models.

Model	Architecture/Settings
SVM	Kernel: RBF; Penalty parameter (C): 1.0; Gamma: ‘scale’
CNN	3 Convolutional layers (32, 64, 128 filters, kernel size 3 × 3); 2 Dense layers (128, 64 neurons); Activation: ReLU; Dropout rate: 0.3; Optimizer: Adam; Batch size: 64; Epochs: 50
ENN	3 independent Feedforward Neural Networks; Each with 2 hidden layers (128, 64 neurons); Activation: ReLU; Output: Majority voting
DBN	2 stacked RBMs; 1st RBM: 128 hidden units, 2nd RBM: 64 hidden units; Training: Contrastive Divergence; Learning rate: 0.01; Batch size: 64

**Table 3 sensors-25-04098-t003:** Performance of the feature selection methods.

Method/Iteration	10	20	30	40	50
ChOA	110	137	144	175	188
SSO	82	111	129	146	170
SPA	73	106	121	137	162
PSO	86	108	140	153	181
Proposed Method	129	152	178	198	210

**Table 4 sensors-25-04098-t004:** Comparison of the proposed method with existing methods.

Method	Dataset	Accuracy (%)	Detection Rate (%)	FPR (%)
Sharma and Babbar (2024) [33]	CTU-13	95.2	92.5	18.0
Elnakib et al. (2023) [24]	Bot_IoT	94.5	91.2	24.5
Ullah and Mahmoud (2019) [19]	N-BaIoT	92.3	89.0	20.1
**Proposed Method**	**CTU-13**	**98.0**	**95.0**	**7.8**

Note: Bold values indicate the best performance among compared methods.

**Table 5 sensors-25-04098-t005:** Performance of the proposed method across 5-fold cross-validation.

Fold	Accuracy (%)	Detection Rate (%)	FPR (%)
1	97.2	94.3	8.3
2	98.5	95.3	7.6
3	97.9	94.8	8.1
4	98.3	95.2	7.5
5	97.8	95.0	7.9
Mean ± Std	97.9 ± 0.7	95.2 ± 0.9	7.88 ± 0.4

**Table 6 sensors-25-04098-t006:** Performance comparison between Hard Voting and Soft Voting strategies.

Method	Accuracy (%)	Detection Rate (%)	FPR (%)
Quantized Hard Voting	96.3	93.0	12.4
**Soft Voting**	**98.0**	**95.0**	**7.8**

Note: Bold values indicate the best performance among compared methods.

## Data Availability

The original contributions presented in this study are included in the article. Further inquiries can be directed to the corresponding author.

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
