# Peer review of "Securing IoT Communications via Anomaly Traffic Detection: Synergy of Genetic Algorithm and Ensemble Method"

_sensors, 2025, doi:10.3390/s25134098_

Round 1
Reviewer 1 Report
Comments and Suggestions for Authors The paper proposes an IoT anomaly detection framework. My major comments are as follows
- The paper need to clarify does it use dragonfly algorithm or simulated annealing?
- What is the other model in table 2?
- Which metaheuristic is used for hyperparameter tuning and feature selection?
- Is the dataset balanced or not? And how many is the total number of sample in the training data?
- Several different baselines like SVM and CNN is included, but no detail of those baselines’ architectures, hyperparameters is included.
Author Response
Comment #1) The paper need to clarify does it use dragonfly algorithm or simulated annealing?
Authors Response) Thank you for your valuable comment. In our work, we have employed Simulated Annealing (SA) for hyperparameter tuning. However, to enhance the exploratory behavior of SA, we have inspired certain aspects of the dragonfly swarm behavior (such as alignment, cohesion, and separation). We clarify that the full Dragonfly Algorithm (DA) was not used in this paper. Only specific behavioral patterns from dragonflies were incorporated to improve SA's search capabilities. We have revised the manuscript to explicitly clarify this point in the "Hyperparameter Tuning" section (Section 3.4).
Comment #2) What is the other model in table 2?
Authors Response) Thank you for highlighting this point. We apologize for the lack of clarity regarding the methods compared in Table 2. To address your comment, we have added brief descriptions of the models used.
Comment #3) Which metaheuristic is used for hyperparameter tuning and feature selection?
Authors Response) Thank you for your valuable comment. In our study, we utilized two different metaheuristic algorithms for different tasks:
- For feature selection, we employed a Genetic Algorithm (GA) enhanced with an eagle-inspired search mechanism to improve exploration and convergence speed.
- For hyperparameter tuning, we used Simulated Annealing (SA), which was further improved by integrating dragonfly-inspired swarm behavioral factors such as alignment, cohesion, and separation.
We have revised the manuscript to clearly highlight these points in the "Materials and Methods" section to eliminate any ambiguity.
Comment #4) Is the dataset balanced or not? And how many is the total number of sample in the training data?
Authors Response) Thank you for your insightful comment. Regarding the class distribution, the dataset is imbalanced, with 4,775 botnet samples and 20,902 normal samples. We have clarified this point in the revised manuscript.
Comment #5) Several different baselines like SVM and CNN is included, but no detail of those baselines’ architectures, hyperparameters is included.
Authors Response) Thank you for your important observation. In response, we have added a detailed description of the baseline models' architectures and hyperparameter settings used in our experiments.
Again, we thank to the reviewer for his timely review and constructive feedback, which have added strength and quality to our manuscript. We hope that we have satisfactorily addressed the reviewer’s comments.

Reviewer 2 Report
Comments and Suggestions for Authors
The article addressed anomaly detection in IoT communications using a machine learning-based multi-phase approach. While the article is well-written and proposes some interesting ideas, the authors need to address the following issues before it can be published.
- The related works section summarises existing literature, however, the discussion is disjointed. There is no natural flow and it is hard to understand the contribution/gaps in existing literature. The authors have provided a table to highlight the disadvantages of existing work, but it is not quite clear how their current work can address some or all of them. The authors should also consider grouping similar works from existing literature together and highlight the gap in existing literature. This will provide a foundation for their existing work. Currently, there is no conclusion from the related work section to highlight the need for their proposed work.
- The authors need to describe the dataset sufficiently. Currently, they are referring to an existing paper but it would be beneficial for the reader if the authors could highlight different aspects of the dataset and justify why they have selected this particular dataset.
- The authors mentioned addressing null values and missing values in phase 1 but did not clearly explain how they addressed this for this dataset.
- The authors compared their work with a few classifiers, however,r they did not show any comparison with existing works using a similar dataset. Are there any existing works on Botnet detection where they used the same dataset? If not, the authors should compare their work with other works on Botnet detection.
- The authors mentioned that they used two types of voting, however, they did not show a comparison of these two models. It is not quite clear which voting type represents their proposed method.
- The authors should also highlight how they separated training and testing datasets, how they created datasets with different attack percentages and their handling of imbalanced data.
- Minor issues:
- The authors should ensure that the full form of the words is presented before using the abbreviated words (e.g., SPA, SSO, etc.)
Author Response
Reviewer #2 The article addressed anomaly detection in IoT communications using a machine learning-based multi-phase approach. While the article is well-written and proposes some interesting ideas, the authors need to address the following issues before it can be published.
Author Response) We thank the reviewer for the positive comment. We used yellow text
Highlight color for the second reviewer.
Comment #1) The related works section summarises existing literature, however, the discussion is disjointed. There is no natural flow and it is hard to understand the contribution/gaps in existing literature. The authors have provided a table to highlight the disadvantages of existing work, but it is not quite clear how their current work can address some or all of them. The authors should also consider grouping similar works from existing literature together and highlight the gap in existing literature. This will provide a foundation for their existing work. Currently, there is no conclusion from the related work section to highlight the need for their proposed work.
Authors Response) We would like to sincerely thank the reviewer for their valuable and constructive feedback. In the original version of the manuscript, we chose not to group the reviewed studies separately because all the related works focused on machine learning-based intrusion detection for IoT networks. As a result, we presented them together to maintain consistency. However, following the reviewer’s helpful suggestion, we have revised the Related Work section by explicitly highlighting the existing research gaps after discussing the prior works. We have now clearly pointed out the limitations of previous studies, such as high computational complexity, lack of scalability, and insufficient adaptability to dynamic IoT environments. These identified gaps now lay a clearer foundation for motivating our proposed framework.
Comment #2) The authors need to describe the dataset sufficiently. Currently, they are referring to an existing paper but it would be beneficial for the reader if the authors could highlight different aspects of the dataset and justify why they have selected this particular dataset.
Authors Response) We thank the reviewer for their valuable suggestion. Following the recommendation, we have revised the manuscript to include a detailed description of the CTU-13 dataset.
Comment #3) The authors mentioned addressing null values and missing values in phase 1 but did not clearly explain how they addressed this for this dataset.
Authors Response) We appreciate the reviewer’s comment. In the revised manuscript, we have clarified the specific methods employed to handle missing and null values in the CTU-13 dataset (Phase 1).
Comment #4) The authors compared their work with a few classifiers, however, they did not show any comparison with existing works using a similar dataset. Are there any existing works on Botnet detection where they used the same dataset? If not, the authors should compare their work with other works on Botnet detection.
Authors Response) We thank the reviewer for the important comment. In response, we conducted an extended literature review and identified several works related to botnet detection using the CTU-13 dataset and similar datasets.
Comment #5) The authors mentioned that they used two types of voting, however, they did not show a comparison of these two models. It is not quite clear which voting type represents their proposed method.
Authors Response) We thank the reviewer for highlighting this important point. In the revised manuscript, we have clarified that we implemented both Hard Voting and Soft Voting ensemble strategies during experimentation. After comparative analysis, it was observed that Soft Voting produced slightly better overall performance in terms of accuracy and detection rate. Therefore, the Soft Voting approach was selected as the final voting mechanism for the proposed method. We have added a comparative table showing the performance results of both voting strategies, and explicitly stated that Soft Voting is used in the final framework.
Comment #6) The authors should also highlight how they separated training and testing datasets, how they created datasets with different attack percentages and their handling of imbalanced data.
Authors Response) We thank the reviewer for the important observation. In the revised manuscript, we have provided additional details regarding dataset preparation and handling.
Comment #7) The authors should ensure that the full form of the words is presented before using the abbreviated words (e.g., SPA, SSO, etc.).
Authors Response) We appreciate the reviewer’s careful observation. In the revised manuscript, we have ensured that all abbreviations, such as Sparrow Search Algorithm (SPA), Social Spider Optimization (SSO), Particle Swarm Optimization (PSO), and others, are introduced with their full forms at their first appearance.
Again, we thank to the reviewer for his timely review and constructive feedback, which have added strength and quality to our manuscript. We hope that we have satisfactorily addressed the reviewer’s comments.

Round 2
Reviewer 1 Report
Comments and Suggestions for Authors Thanks for the revision. The quality of manuscript is improved. Though I have few comments could further improve the paper.
- the paper needs to better explains why the specific GA ensemble design outperform alternatives baselines.
- What is the practical implication of the proposed model? And I suggest to include the section to discuss the strength and limitation of the models.
- I suggest to include the recent work of GNN for traffic anomaly detection. For instance "End-to-end heterogeneous graph neural networks for traffic assignment" and "Graph neural network surrogate for seismic reliability analysis of highway bridge systems"
- Table 3’s objective function is not clear. What is the metric you are comparing? Furthermore, I suggest to include k-fold cross validation and report the variance across runs
Author Response
Reviewer #1 The paper proposes an IoT anomaly detection framework. My major comments are as follows:
Authors Response #1 ) We thank the reviewer for the positive and valuable comments. We used Bright Green text highlight color for the first reviewer.
Comment #1) The paper needs to better explains why the specific GA ensemble design outperform alternatives baselines.
Authors Response #2) We thank the reviewer for this insightful comment. To address this, we have added a detailed explanation in the revised manuscript discussing why the proposed GA-ensemble framework achieves superior performance compared to the baselines (SVM, CNN, ENN, DBN). Specifically, the proposed system benefits from three key innovations:
- Optimal Feature Selection with GA and Eagle-Inspired Search: Unlike standard optimization methods, our GA is enhanced by eagle-inspired spiral search, which improves exploration of the feature space and avoids local optima. This results in a more relevant and compact feature subset, which in turn improves classifier performance and reduces computational cost.
- Swarm-Based Hyperparameter Tuning: The use of a simulated annealing algorithm inspired by dragonfly swarm behavior enables fine-grained tuning of the ensemble classifiers, enhancing their individual performance and synergy.
- Ensemble with Diverse and Complementary Learners: The combination of Decision Tree, Random Forest, and XGBoost provides both bias-variance trade-off and resilience to imbalanced data.
Comment #2) What is the practical implication of the proposed model? And I suggest to include the section to discuss the strength and limitation of the models.
Authors Response) We thank the reviewer for the valuable suggestion. In response, we have incorporated a new paragraph at the end of Section 4.2 (Discussion) that highlights the practical implications, strengths, and limitations of the proposed model. This addition explains how the framework can be applied to real-world IoT applications, including smart cities, healthcare monitoring systems, and industrial control networks especially in environments that demand lightweight, accurate, and scalable anomaly detection solutions. Furthermore, the discussion emphasizes the model's major strengths, such as its high detection accuracy, low false positive rate, and adaptability to diverse network conditions. It also acknowledges key limitations: potential performance degradation when handling encrypted or obfuscated traffic, and the computational overhead introduced during the automated hyperparameter tuning phase.
Comment #3) I suggest to include the recent work of GNN for traffic anomaly detection. For instance "End-to-end heterogeneous graph neural networks for traffic assignment" and "Graph neural network surrogate for seismic reliability analysis of highway bridge systems".
Authors Response) We appreciate the reviewer’s suggestion to include recent developments in GNN-based traffic analysis. In response, we have added a short paragraph in the Related Work section to briefly review these recent advances. Specifically, we now reference Liu and Meidani (2024) on end-to-end heterogeneous GNNs for traffic assignment, which leverage attention mechanisms and flow conservation laws to improve prediction accuracy, and their GNN surrogate model for seismic reliability analysis of bridge networks, which showcases the potential of GNNs in learning graph-based dependencies in transportation systems. While our work focuses on ensemble learning rather than graph-based learning, we acknowledge these GNN approaches as promising directions for future research in IoT-based anomaly detection.
Comment #4) Table 3's objective function is not clear. What is the metric you are comparing? Furthermore; I suggest to include k-fold cross validation and report the variance across runs.
Authors Response) We thank the reviewer for raising this important point. In the revised manuscript, we have clarified the objective function and evaluation metrics used in Table 3.
Additionally, we have incorporated a 5-fold cross-validation procedure to enhance the robustness of the evaluation. We now report both the mean and standard deviation of the performance metrics across the five folds.
************
Again, we thank the reviewers for their timely review and constructive feedback, which have added strength and quality to our manuscript. We hope that we have satisfactorily addressed the reviewers’ comments.

Reviewer 2 Report
Comments and Suggestions for Authors
I am happy with the revisions made by the authors and would recommend for publication.
Author Response
Responses to the Reviewer 2 Comments
Reviewer #2 I am happy with the revisions made by the authors and would recommend for publication.
Authors Response ) We thank the reviewer for the positive evaluation of our work
